# Impact of Trump's Digital Rhetoric on the US Elections: A View from Worldwide Far-Right Populism

**Concha Pérez-Curiel, Rubén Rivas-de-Roca \* and Mar García-Gordillo**

Department of Journalism II, University of Seville, Avda. Américo Vespucio, s/n., 41092 Seville, Spain; cperez1@us.es (C.P.-C.); marggordillo@us.es (M.G.-G.)

\* Correspondence: rrivasderoca@us.es

**Abstract:** A time of turmoil and uncertainty is invading the public sphere. Under the framework of the 2020 US elections, populist leaders around the world supported Trump's speech on Twitter, sharing a common ideology and language. This study examines which issues (issue frame), and strategies (game frame) framed the messages of populism on Twitter by analyzing the equivalences through Trump's storytelling and checking the bias of the media in the coverage of the US elections. We selected a sample of tweets (*n* = 1497) and digital front pages of global newspapers (*n* = 112) from the date of the Trump/Biden face-to-face debate (29 September 2020) until the Democratic party candidate was proclaimed the winner of the elections by the media (7 November 2020). Using a content analysis method based on triangulation (quantitative and qualitative-discursive), we analyzed the Twitter accounts of five leaders (@realDonalTrump, @MLP_officiel, @matteosalvinimi, @Santi_ABASCAL, and @Jairbolsonaro) and five digital front pages (*The New York Times*, *O Globo*, *Le Monde*, *La Repubblica,* and *El País*). The results show that populist politicians reproduced the discourse of fraud and conspiracy typical of Trump's politics on Twitter. The negative bias of the media was also confirmed, giving prominence to a rhetoric of disinformation that overlaps with the theory of populism.

**Keywords:** political populism; Trump; Twitter; elections; United Stated; polarization; disinformation; legacy media; voters

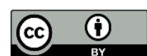

## 1. Introduction

The impact of the last United States (US) presidential elections on world geopolitics were unprecedented. The face-to-face debate between Donald Trump and Joe Biden as candidates was the starting point of a campaign marked by polarization and conflict (Neudert and Marchal 2019). The assault on the Capitol (January 2021), carried out by extremist groups linked to the Republican party (The New York Times 2021), and the judicial impeachment process against Trump (February 2021) are consequences of populist rhetoric that mobilized citizens through social networks.

In the context of a global institutional crisis of democracy (Levitsky and Ziblatt 2018), reinforced by the breakthrough of COVID-19, some authors sought to find causes in factors such as platformization (Smyrnaios and Rebillard 2019) or the lack of regularization of data brokers. Algorithms and bots are used for manipulation and computational propaganda (Woolley and Howard 2017). This practice fosters the uncontrolled dissemination of fake news in political processes (Powers and Kounalakis 2017). In line with this premise, social media contribute to the spread of disinformation (Rivas-de-Roca et al. 2020), but it is also necessary to consider other variables, such as the influence of populism, assessed as a persuasion tool through language (Fuentes Rodríguez 2020).

The 2016 electoral campaign in the US already provided evidence on Trump's effectiveness at carrying out storytelling through Twitter-based strategies of far-right

populism (Pérez-Curiel and Limón-Naharro 2019). He developed an opposite position to globalization, integration, and establishment policies (Mudde 2016), supranational entities such as the European Union (Mammone 2009), and immigrants, refugees, and the Muslim culture (Wodak 2015; Fuchs 2017). In the 2020 elections, leaders of populist parties in Europe and around the world retweeted Trump's messages on their Twitter accounts. Marine Le Pen (France), Matteo Salvini (Italy), Santiago Abascal (Spain), and Jair Bolsonaro (Brazil) concurred with Trump's theories of conspiracy and fraud (Fajardo-Trigueros and Rivas-de-Roca 2020), denying the legitimacy of the election results. Some of the leaders belong to different political families, but they have in common an aggressive rhetoric defending the interests of the "people" against the elite (Acemoglu et al. 2013).

A scenario where there is a great level of disinformation was generated. International organizations (European Commission 2018), together with social platforms (Facebook, Google, and Twitter), warned of a problem which, in 2018, was considered a threat to democracy by 83% of Europeans. They were highly concerned about the increase in online fake news during electoral periods (EUvsDisinfo 2018). As public opinion polls confirm, "6 out of 10 Americans believe that Biden legitimately won the election. But 7 out of 10 Republicans affirm that he was not legitimately elected" (Washington Post-ABC 2020). This is an indicator of distrust of voters towards institutions (Waisbord 2018).

As an open research question, we seek to know the narratives used by populist leaders. Bearing this in mind, it is also essential to check the practices of the media. Far from fighting disinformation, press coverage showed elements of fallacy and propaganda typical of far-right populism (Carlson 2017; Bennett and Pfetsch 2018), specifically by applying a critical bias to politicians. The objective of this study was to determine the impact and influence of Trump's speech during the US elections on the leaders of extreme right-wing populism countries, analyzing their issues (issue frame), strategies (game frame), and rhetorical marks of disinformation. We also explore the information bias of the press in each country based on the selection of topics and the journalistic treatment of tweets published by the leaders. The focus on right-ring populism is based on its huge use of social media and the impact on the public sphere (Bimber and Gil de Zúñiga 2020).

## 2. Theoretical Framework

### 2.1. Pro-Trump Populism. Leader Digital Rhetoric Supporting the Fraud

The rise of populism in Europe and Latin America has occurred as a consequence of the crisis of liberalism (Nye 2017). Representative democracy may face a serious risk of decline. This idea is endorsed by the increasing level of support for authoritarian regimes. Citizens do not consider the democratic system positive or desirable (Foa and Mounk 2016). The weakness of parliamentarism (Kelsen 2005) highlights the inability to react to the effects of extreme populism at the political level (The Guardian 2019) with a relevant presence of the most conservative positions (Bevelander and Wodak 2019).

Against this backdrop, European extreme right-wing populisms, represented by Matteo Salvini (Northern League), Marine Le Pen (National Rally), and Santiago Abascal (Vox), are examples of success in both the national and European elections. Their policies do not identify with traditional populism but with the so-called "post-industrial" populism, which is not linked to fascist positions, being proponents of a new agenda (Ignazi 2006). Some key characteristics of these leaders are that they have xenophobic, protectionist, and nationalist values as well as criticism of traditional elites. Indeed, the Le Pen phenomenon happened before Trump's victory in 2016, showing the early consolidation of populism in Europe. Regarding Latin America, the rise of violence, impunity, and corruption interfere with social order and create a feeling of insecurity among citizens (Serrano Rodríguez 2019). The rise of populist policies is fueled by a time of uncertainty and growing distrust in traditional politics (Acemoglu et al. 2013).

Specifically, in the US, a climate of polarization (Graham et al. 2013), pop-politics (Baym 2010), and social chaos (Waisbord 2018) existed from the start of the 2020 campaign.

There was a macro-strategist leader who undermined the legitimacy of traditional parties and governments and promoted disinformation on social media. As in the 2016 presidential elections, Trump stood out for his constant appeal to emotional feelings, xenophobic statements against minorities (Fuchs 2017), and nationalist domestic and foreign policies (Ramírez Nárdiz 2020).

Behind his simple and repetitive language, a strategy of fraud emerges. Trump represents worldwide populism defined by a first-person narrative. This can be seen in discursive antagonism shared against the other, conspiracy theories, and an emphasis on the homeland (*America First*). Populist leaders draw a convergent line with Donald Trump's strategies and language. A sizable body of literature has problematized the promotion of antipluralism (De la Torre 2010) and the management of public emotions (Beckett and Deuze 2016) related to this global trend. These populist politicians have criticized the legacy media and labeled them as antagonists (Waisbord and Amado 2017), eroding independent journalism and democracy (Pérez-Curiel 2020). These populist politicians have used social media channels like Twitter to criticize the contents of legacy media.

Twitter is a key social network of non-mediated communication that allows direct contact with people and avoids the traditional media, who are labeled as conventional "elites" (van Kessel and Castelein 2016). In this sense, an alternative non-mediated agenda is developed (Enli 2017), increasing interaction with citizens (Rúas Araújo et al. 2018). Populist leaders focus more on opinions than facts, making extensive use of the cyber-rhetoric for the purpose of achieving votes (Stromer-Galley 2014). This practice comes from populist theory and undermines the political establishment (Engesser et al. 2017), mobilizing citizens against the system (Crilley and Gillespie 2019). However, populism with decision-making power is different from the populist leaders that try to achieve parliamentary representation. The aggressiveness of the rhetoric is higher in the latter, having both a permanent-campaign style (Maatsch 2021).

Interestingly, Twitter also boosts fake news and the spreading of hate speech (Bimber and Gil de Zúñiga 2020; Bracciale and Martella 2017). Individuals take advantage of social networks to promote machismo, homophobia, xenophobia, Islamophobia, anti-Semitism, and other forms of intolerance, making them seem acceptable (Colleoni et al. 2014). These strategies intensify the spread of disinformation, propaganda, and hoaxes (Salaverría et al. 2020), opening a debate on the role of the media in verifying facts.

*2.2. US Election Narrative in the Media. Convergence with Far-Right Populism*

In a context affected by COVID-19, citizens are facing a social situation of chaos, anxiety, and confusion, which has increased interest in consuming news through social media (Newman et al. 2020). At the same time, the public's distrust of elites and the media is a political trend (Shearer and Gottfried 2017). This phenomenon is also associated with the growth of alternative sources of information linked to populism and the far-right movements (Bennett and Livingston 2018).

The role of the media in the coverage of the US elections validates the principles of the first- and second-level agenda-setting theory (McCombs 2005). The media decide the issues and also evaluate the substantive dimension (ideology, position of the candidate, qualification, and personality) and affective dimension (positive, neutral, or critical opinion of the facts). At electoral stages, they prioritize news related to a candidate's strategies (game frame) over the topics of the political program (issue frame). Furthermore, there is a growing level of personalization, progressive negativism, and a news narrative related to feelings ahead of rational argumentation (Marzal-Felici and Casero-Ripollés 2017).

As previously described, the main candidates of far-right international populism broke into the social networks supporting Trump's denialist and conspiracy theories. They alluded to immigration, foreign affairs, and environmental or gender issues in a context marked by the pandemic. Along with this, the websites of media outlets such as *Le Monde*, *O Globo*, *La Repubblica*, and *El País* devoted more space to issues related to the Trump fallacy than to the rest of the news. Thus, these media deviated from reporting

responsibly and carrying out their role as generators of public opinion (Casero-Ripollés et al. 2017). The national and international press acted in a double sense: it either presented a provocative, offensive, and uncivil discourse around Trump and the populist leaders on Twitter (Ott 2017) or refuted disinformation with well-contrasted news (Mantzarlis 2018; Vázquez-Herrero et al. 2019).

Moreover, information bias in the newspapers should be kept in mind. Leaders' confrontation with the media is a characteristic of populist policies, especially in the case of Donald Trump. In the 2016 campaign, the digital front pages of *USA Today*, *The Boston Globe*, *The Wall Street Journal*, and *The New York Times* already showed indications of the attitude of a sector of the Republican and Democratic media towards this political figure and his discourse of infotainment (Pérez-Curiel and Limón-Naharro 2019).

Our research aims to investigate the issues and brands related to the populist discourse based on an analysis of Twitter and media coverage. To do this, we pose three research questions:

**RQ1:** *Which themes and strategies of far-right populism can be identified in the speech of Trump and other populist leaders on Twitter?*

**RQ2:** *Is disinformation a characteristic of the messages posted by politicians on Twitter?*

**RQ3:** *Do the media show an information bias in the treatment of the issues published by populist leaders on Twitter?*

### 3. Method

*3.1. Sample Procedure*

Data from this study were obtained using a triangulated method. A quantitative and qualitative-discursive content analysis (Bardin 1977; Manfredi et al. 2021) was applied to the tweets of political leaders and the front pages of newspapers. The reason for this multimodal approach lies in the consolidation of a hybrid communication model between digital and legacy media (Chadwick 2017), including Twitter as a source for the media system (Justel-Vázquez et al. 2018; Hermida and Mellado 2020).

Populist politicians develop their own style in social networks to spread their ideologies in a fragmented way (Block and Negrine 2017; Engesser et al. 2017) based on the identification of enemies, such as migrants, who must be fought (van Kessel and Castelein 2016; Arcila-Calderón et al. 2020). Consequently, our methodological proposal was designed to analyze the Twitter profiles of Donald Trump and four others widely recognized populist leaders, making it possible to compare America and Europe.

We focused on Twitter because of its advantages for political communication in elections (Gainous and Wagner 2014; D'Heer and Verdegem 2015). Indeed, the 2008 and 2012 US elections showed interesting benefits of the Internet in terms of the stability of democracy, which were put into question by the use of Twitter in 2016 (Campos-Domínguez 2017). Taking into account these milestones, the details of the sample of populist profiles on Twitter are now exposed:

- Donald Trump (United States), @realDonaldTrump (unique account in English);
- Jair Bolsonaro (Brazil), @jairbolsonaro (unique account in Portuguese);
- Marine Le Pen (France), @MLP_officiel (unique account in French);
- Matteo Salvini (Italy), @matteosalvinimi (unique account in Italian);
- Santiago Abascal (Spain), @Santi_ABASCAL (unique account in Spanish).

In addition to the issues and discursive elements of the messages published by these politicians on Twitter, our analysis considered their concordance with topics on the front pages of newspapers, since recent research has outlined the impacts of tweets in traditional media (Rúas Araújo et al. 2018; Pérez-Curiel and Limón-Naharro 2019).

Furthermore, the polarization on Twitter around controversial topics for right-wing populism, such as climate change, could influence public opinion (Moernaut et al. 2020).

Hence, a newspaper was selected for each country, considering circulation rates according to data from the World Association of Newspapers and News Publishers (WAN-IFRA). The sample was composed of *The New York Times* (United States), *O Globo* (Brazil), *Le Monde* (France), *La Repubblica* (Italy), and *El País* (Spain). These were used to conduct a comparative analysis of different media systems, and the focus was expanded by including Brazil, which is outside the Western world, as suggested by the authors of the theory (Hallin and Mancini 2017).

To assess the similarities and differences between Trump's speech and the narrative of the populist leaders, we used a data collection period from 29 September to 8 November 2020, that is, 40 days. The reason for setting 29 September as the beginning is that, on that date, the first TV electoral debate between Biden and Trump was held, marking the beginning of the campaign. The end date refers to the day after the winner of the elections was known, which allowed us to consider possible reactions (Rivas-de-Roca et al. 2020). Post-electoral surveys from traditional media outlets, such as CNN (2020) and NBC (2020), showed a great thematic division of US voters on major national issues. It was therefore of interest to investigate the reactions of citizens to the results.

The sample included all of the tweets published by the leaders selected during the defined electoral period, as well as the front pages of national newspapers in which information about these elections appeared. This research studied their own tweets and the candidates' responses, but not the retweets, since they included information published by other sources that were not necessarily linked to the agenda of each leader (Larsson and Ihlen 2015; Casero-Ripollés et al. 2017). The sample was captured through Twitonomy and the websites of the newspapers and was subsequently analyzed with SPSS statistical software. In total, 1497 publications on Twitter and 112 front pages in the digital press were collected.

*3.2. Issue and Game Frame Variables: Issues/Strategies and Propaganda Mechanisms*

Our research sought to identify the interactions between politicians, media, and citizens, which are presented as clearly endogamous, tending towards the creation of echo chambers, in the literature (Colleoni et al. 2014; Guerrero-Solé 2018). Populist figures can act as opinion leaders in classical conception (Katz and Lazarsfeld 1955), using identity building, a cyber-rhetoric (López-Meri 2016), and their relationships with the media as principles of action (Block and Negrine 2017). Thus, the following quantitative/qualitative worksheet was developed (Table 1) to join the thematic and strategic content analysis with the use of language:

**Table 1.** Contingence quantitative/qualitative variables.

| Twitter and legacy media | Tweets from populist leaders / Front pages on which these leaders appeared | Evaluative and formal indicators | Theme (issue frame) | Strategy (game frame) | Language (fallacy/propaganda) |
|---|---|---|---|---|---|

Regarding the assessment of the content, a quantitative analysis was used, which allowed us to investigate the items that made up the messages in depth (Neuendorf 2002; Krippendorff 2012). This method has been adapted to social networks such as Twitter by some authors (Fernández Crespo 2014). Our study focused on the thematic agenda (issue frame) and the tools for obtaining votes (game frame) since they are the two principal frames of current political communication (Aalberg et al. 2017; Alonso-Muñoz and Casero-Ripollés 2020).

Therefore, a specific analysis worksheet with exclusive categories was used to analyze the tweets (Table 2). The aforementioned issue frame/game frame theories were considered (Aalberg et al. 2017; Cartwright et al. 2019) to assess the use of agendas and strategies and observe their effects on audiences. The genesis of populist discourse is increasingly being linked to platformization and computational propaganda (Arcila-Calderón et al. 2020).

**Table 2.** Categories used for the quantitative study of the agenda on Twitter.

| | Items | Description |
|---|---|---|
| Issue frame | Conspiracy theories | Tweets regarding possible conspiratorial explanations for social problems, such as those mentioning George Soros. |
| | Immigration/security | Tweets that connect immigration to citizen security issues. |
| | Corruption | Tweets related to malpractices by traditional political authorities. |
| | Gender issues | Tweets on gender issues to criticize equality policies. |
| | COVID-19 | Tweets on the COVID-19 pandemic as a singular matter of public interest. |
| | Environment | Tweets that refer to environmental issues, usually from a denial approach. |
| | Foreign affairs | Tweets on international affairs, such as trade or relations between countries. |
| | Economy | Tweets on economic issues, such as unemployment, subsidies, or industry. |
| Game frame | Horse race and governing frame | Tweets that refer to opposing positions, post-electoral pacts, or government strategies. |
| | Politicians as individuals' frames | Tweets that mention aspects of the personal lives of populist leaders. |
| | Political strategy frame | Tweets on political events, such as electoral debates or meetings with citizens. |
| | News management frame | Tweets related to the media, such as interviews to the candidate or the existence of discrepancies with a journalistic work. |
| Other | Unclassifiable tweet in the previous categories | |

Regarding the discursive analysis (van Dijk 2008), a range of categories on political language was used, applying a classification of fallacies and propaganda mechanisms, as follows:

- Appeal to authority;
- Appeal to emotion;
- Fallacy against the man;
- Appeal to force;
- Appeal to ignorance;
- Attributions;
- Tendentious claims;
- Emphasis;
- Stereotypes;
- False analogy;
- Speaking through other sources;
- Opinions as facts;
- Selecting information;
- Use of labels.

In recent years, the amount of hate speech on the Internet has increased (Bartlett et al. 2014), encouraging studies to delve into the linguistic building of messages (Schmidt and Wiegand 2017). The mentioned categories of fallacies and propaganda mechanisms were studied with a critical discourse analysis (Flowerdew and Richardson 2017), something that has already been used in previous research on populism (Alonso-Muñoz and Casero-Ripollés 2020).

The whole analysis was carried out manually by the authors. We chose IBM SPSS Statistics, Version 25, as the statistical software to process the data. The intercoder agreement was calculated with Scott's Pi formula, reaching an acceptable error level of 0.96. Two previous rounds of coding training, of 5 days each one, were held; meanwhile, control variables were not applicable. The method suits very well for nominal data in communication studies, allowing us to study the agendas and propaganda mechanisms presented in the sample.

## 4. Results

### 4.1. Description of the Sample

The sample used in this research was composed of 1497 tweets, divided as follows: Salvini 845 (56.2% of the total), Trump 237 (15.8%), Bolsonaro 217 (14.5%), Le Pen 131 (8.75%), and Abascal 67 (4.8%). Therefore, the data outline the overactivity of Salvini that is relevant. This means that the figures for the total sample are not very useful, as they over-represent the Italian. Instead, meaningful comparisons can be drawn between the profiles.

As for the front pages, the sample was small (*n* = 112), as expected, but it did allow us to correlate the occurrence of the elections in the United States with populist activity on Twitter. The frequencies were as follows: *New York Times* = 37, *O Globo* = 29, *Le Monde* = 9, *La Repubblica* = 18, and *El País* = 19. It was observed that the great level of activity of Salvini on Twitter did not correspond with his appearance in the newspaper selected from his country (*La Repubblica*), which suggested that other journalistic factors should be considered. However, the reference to the US elections was very common in the 40-day period analyzed, as seen in Figure 1.

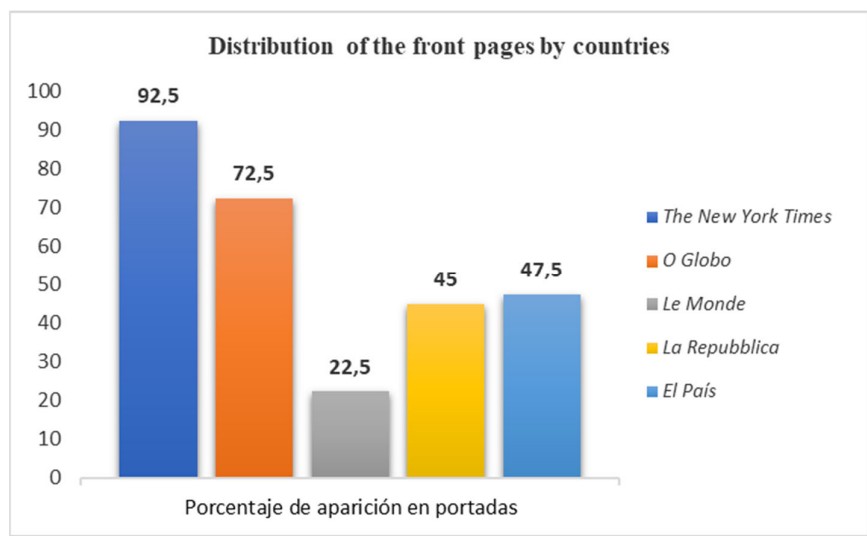

**Figure 1.** Comparison of references to the US elections on the front pages (%).

We find that the *The New York Times* covered the elections on 92.5% of its front pages, which can be explained by the fact that the electoral contest was taking place in this country. It is more surprising that other reference media worldwide gave so much space to this matter. This first finding shows the media relevance of the US campaign in the mainstream press, revealing its usefulness in defining a global populist policy.

### 4.2. Strategy and Propaganda on Twitter

The analysis of the thematic and strategic agendas of these leaders on Twitter provides interesting data that show the similarities and divergences in their communication practices. First, Salvini and Abascal displayed relatively fragmented agendas with some

points in common. Both used a wide variety of topics, with COVID-19 and security and immigration being the most commonly mentioned issues. There was also a plurality of frames in relation to strategies, although the dispute approach (horse race) was the most commonly applied in both cases (Table 3).

**Table 3.** Distribution of tweets according to topics and strategies (%). * The most outstanding figures are presented in bold, since they show relevant trends.

|  |  | Donald Trump | Jair Bolsonaro | Marine Le Pen | Matteo Salvini | Santiago Abascal |
|---|---|---|---|---|---|---|
| Issue frame | Conspiracy theories | 4.7 | 1.9 | - | 2 | 10.4 |
|  | Immigration/security | 3 | 3.2 | **39.7** | **23** | **13.4** |
|  | Corruption | **47.5** | 2.3 | - | 0.5 | 3 |
|  | Gender issues | - | - | 1.5 | 1.2 | - |
|  | COVID-19 | 5.5 | 3.7 | **15.3** | **15** | **16.4** |
|  | Environment | - | 3.2 | 3.1 | 1.4 | - |
|  | Foreign affairs | 0.8 | 5.1 | 13 | 5.4 | 7.5 |
|  | Economy | 0.8 | **18.5** | 10.7 | 7.2 | 1.5 |
| Game frame | Horse race and governing frame | **19.9** | **27.8** | 2.3 | **7.2** | **10.4** |
|  | Politicians as individuals' frames | 1.7 | 1.4 | **9.9** | 7.1 | 7.5 |
|  | Political strategy frame | 5.1 | **23.1** | 1.5 | 7.1 | 6 |
|  | News management frame | 1.7 | 5.6 | 3.1 | 4 | 3 |
|  | Other | 9.3 | 4.2 | - | 18.8 | 20.9 |

The preference for this confrontational setting was also the case for Bolsonaro (27.8%) and Trump (19.9%), and in the North American case, this reflected the conflictive character that the electoral campaign acquired. The results suggest that the horse race approach was a priority element for most of the populist leaders in the sample. The exception was Le Pen, who showed a low level of use of this frame, placing personal issues first in her communication strategy.

The use of game frames, particularly the horse race and governing frame, was revealed as being a common characteristic of global populism. Meanwhile, the use of thematic frames was much more distributed and linked to geographical contexts. It is noteworthy that the three European leaders (Abascal, Salvini, and Le Pen) coincided in prioritizing COVID-19 and security and immigration as topics, which shows their relevance to EU politics. Le Pen again displayed a differential view since she concentrated her agenda more, focusing on immigration and security items (39.7%). This may be due to the importance of these aspects in French public opinion.

Moreover, Trump and Bolsonaro ignored COVID-19, probably because it was an issue that had the potential to harm them as heads of government. Bolsonaro granted a huge amount of space to the economy (18.5%) in an attempt to claim his achievements and face criticism about the management of the pandemic. For his part, Trump focused his thematic speech on corruption (47.5%), that is, on the possibility of an electoral fraud that would modify the results. This idea was central to Trump's actions during his campaign on Twitter, which helps us to understand why a large portion of Republican voters believed in his victory after he lost the election (Washington Post-ABC 2020; Pew Research Center 2020).

Beyond the fact that conflictive frames are commonly used in populist tweets (Figure 2), the use of discursive propaganda mechanisms also seems to be frequent. Appeal to emotion (15.6%) and the presentation of opinions as facts (14%) were the most common practices identified in the sample as a whole. These records show the existence of a narrative based on false messages that seek to manipulate the audience (Table 4).

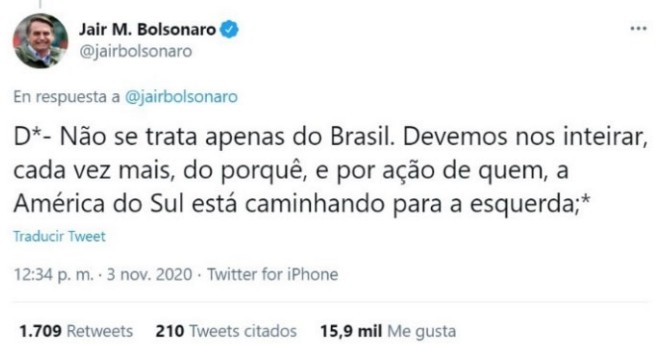

**Figure 2.** Tweet from Bolsonaro using the horse race frame.

**Table 4.** Propaganda mechanisms on Twitter (%). * The most outstanding findings are presented in bold, since they show relevant trends.

|  | Donald Trump | Jair Bolsonaro | Marine Le Pen | Matteo Salvini | Santiago Abascal | Total |
|---|---|---|---|---|---|---|
| Appeal to authority | 1.7 | 3.2 | 3.1 | 5.2 | 6 | 4.2 |
| Appeal to emotion | **15.2** | 6.9 | **49.6** | **13.2** | 9 | **15.6** |
| Fallacy against the man | 0.8 | 3.2 | - | 2.2 | 9 | 2.3 |
| Appeal to force | 6.3 | 0.5 | 2.3 | 7 | **17.9** | 6 |
| Appeal to ignorance | 4.6 | 2.8 | - | 1.2 | 6 | 2.1 |
| Attributions | 2.5 | **29.6** | 0.8 | 8.9 | **10.4** | 10.2 |
| Tendentious claims | 2.5 | 2.8 | - | 0.5 | 3 | 1.2 |
| Emphasis | 5.1 | 0.5 | 14.5 | **13.8** | 7.5 | 10.3 |
| Stereotypes | - | 0.5 | 1.5 | 3.6 | 1.5 | 2.3 |
| False analogy | 3 | - | - | 3 | 4.5 | 2.3 |
| Speaking through other sources | 11.8 | 4.2 | - | 11.8 | - | 9.2 |
| Opinions as facts | 11.8 | **38.9** | 0.8 | 5.9 | **10.4** | **14** |
| Selecting information | **16** | 6 | 27.5 | 7 | 9 | 10.2 |
| Use of labels | 1.7 | - | - | 6.6 | 6 | 4.3 |
| Other | - | 0.9 | - | 10.1 | - | 5.8 |

For Trump, the aforementioned practices were complemented by information selection (16%) and speaking through other sources (11.8%). Thus, there was evidence of manipulation of messages using biased data based on others to criticize competitors. Trump's actions were quite similar to the rest of the populist leaders since he placed great importance on the appeal to emotion (15.2%) and opinions as facts (11.8%). This implies that he spread clearly false tweets, such as those launched after the elections denouncing electoral fraud without any type of proof (Figure 3).

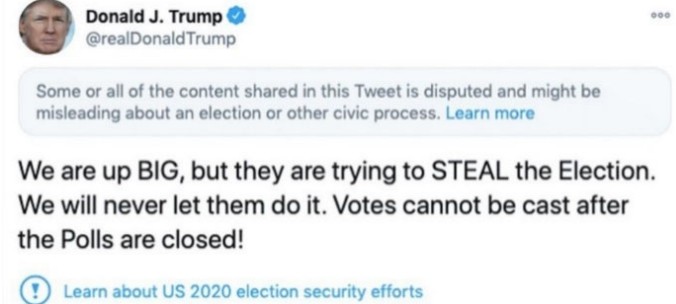

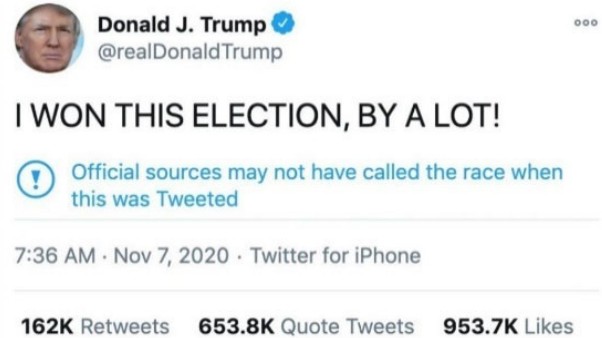

**Figure 3.** Tweets from Trump appealing to emotion and stating opinions as facts.

The other populist leaders also showed interesting divergences, although most of them shared the application of emotions and opinions. For instance, Bolsonaro frequently presented opinions as facts (38.9%) as well as attributions (29.6%). The latter are common in game frame approaches since they emphasize the successes and failures of political actors. Le Pen showed a strong preference for information selection (27.5%) and appealing to emotion (49.6%). These two practices were also carried out by Trump. It must be noted that Le Pen is the only leader who rarely published opinion-based messages as factual (0.8%).

With regard to Salvini, he used many propaganda resources involving all of the mechanisms mentioned. Within this fragmented strategy, the Italian leader prioritized emphasis (13.8%) and classic appeal to emotion (13.2%). In contrast, Abascal was the politician in this research who presented differential behavior. Appeal to force (17.9%), typical of the militaristic environment that surrounds his party, and attributions (10.4%), together with presenting opinions as facts, were found to be his preferred tools.

The sample of tweets analyzed showed the spectacularizing message of populist leaders, which was reinforced many times with propaganda mechanisms. Indeed, the tweets from Trump, Le Pen, and Abascal always used this kind of strategy. Falsehood and emotion were found to work as the basis of these discourses in a common pattern, regardless of national differences. However, these national contexts are relevant to the understanding of adaptions of populism. The preference for opinions instead of facts harms the value of information in a democracy.

### 4.3. Impact of Populism on Legacy Media

At this point, it is interesting to consider how populist strategies are presented in traditional media. The data show that propaganda resources appeared on the front pages of newspapers (Table 5), although there was not a perfect correlation between the language marks prioritized by politicians on Twitter and those picked up in the media. Meanwhile, appeal to emotion (234 mentions), and the presentation of opinions as facts (210) were the mechanisms preferred by leaders; attributions (16.4%) and information selection (14.7%) were the most commonly used strategies on the front pages.

**Table 5.** Propaganda mechanisms. Frequencies of use in tweets/on front pages (%).

|  | **No. of Tweets** | **% of Mentions in Front Pages** |
|---|---|---|
| Appeal to authority | 63 | 6.9% |
| Appeal to emotion | 234 | 5.2% |
| Fallacy against the man | 34 | 5.2% |
| Appeal to force | 90 | 6.9% |
| Appeal to ignorance | 31 | 2.6% |
| Attributions | 153 | **16.4%** |
| Tendentious claims | 18 | 6% |

| | | |
|---|---|---|
| Emphasis | 154 | 6.9% |
| Stereotypes | 34 | 5.2% |
| False analogy | 35 | 1.7% |
| Speaking through other sources | 137 | 0.9% |
| Opinions as facts | 210 | 6.9% |
| Selecting information | 152 | **14.7%** |
| Use of labels | 64 | 6% |
| Other | 87 | 8.6% |

As can be seen from Table 5, the media prefer to use conflictive approaches such as attributions, which pose a direct confrontation between political actors. In addition, the use of information selection was remarkable, particularly the use of data biased by emotion (5.2%) or opinion (6.9%), which are typical false messages. Although propaganda mechanisms were not fully transferred to the front pages, it is worth emphasizing the great presence of these biased resources in the media (only 8.6% of the front pages lack them), showing the journalistic weight of disinformation in the coverage of the US elections.

Nevertheless, in some cases, the tweets published by populist leaders during the US campaign were directly reflected on the front pages (Table 6). The *New York Times*, as the selected media outlet with a high level of reporting on the elections, showed a clear negative information bias (81.7% of the front pages). In other less representative examples, reference to these tweets was either negative (*Le Monde*) or more positive than negative (*O Globo* and *La Repubblica*). However, on average, the analysis of front pages with the presence of tweets reveals a prevalent negative tone (80.4%).

**Table 6.** Frequencies of tweets on the front pages of different media outlets and message tone (%).

| | | Positive | Negative | Neutral |
|---|---|---|---|---|
| Tweets on the front page | *The New York Times* | 17.9 | 81.7 | 0.4 |
| | *O Globo* | 66.7 | - | 33.3 |
| | *Le Monde* | - | 100 | - |
| | *La Repubblica* | 100 | - | - |
| | *El País* | - | - | - |
| | Total | 18.8 | 80.4 | 0.8 |
| No tweets on the front page | *The New York Times* | - | - | 100 |
| | *O Globo* | 47.9 | 23 | 29.1 |
| | *Le Monde* | 11.5 | 86.2 | 2.3 |
| | *La Repubblica* | 17.8 | 64.3 | 17.9 |
| | *El País* | 16.4 | 74.6 | 9 |
| | Total | 22.1 | 60 | 17.8 |

A negative approach (60%) was also identified on front pages in which there were no tweets, despite the higher levels of positive (22.1%) and neutral (17.8%) contents. All of these figures show that negative frames were a constant feature of the journalistic treatment of the US elections, especially when the front pages of newspapers were based on populist tweets.

As we previously noted, the negative approach was identified as a priority in *The New York Times*. During the 2020 election campaign, this prestigious media placed messages from Trump that had been broadcasted primarily on social networks at a top position. This was the case when Trump fostered distrust in the vote-counting process and minimized the real impact of the COVID-19 virus after leaving the hospital (Figure 4).

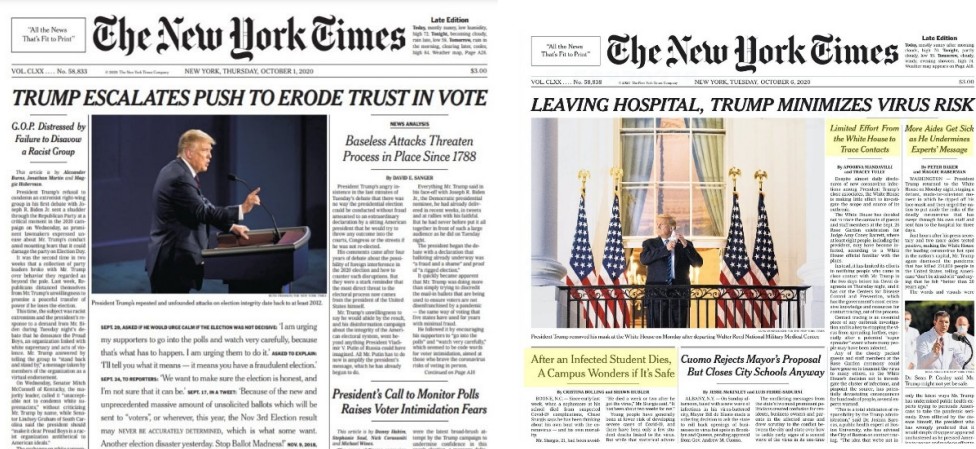

**Figure 4.** Front pages of *The New York Times* in October 2020, replicating tweets from Trump.

The link between traditional media and the messages of a populist leader on a social network provides evidence of the hybridization of the current political landscape. Trump used Twitter as his main communication channel, but the propaganda mechanisms did not remain on social media; rather, they moved to the quality press. Most of the front pages of the sample applied this type of propaganda label, which should trigger a deep reflection on the amplified role of legacy media in far-right populism.

## 5. Discussion and Conclusions

The 2020 US presidential election attracted global attention. Trump's political campaign, the use of a cyber-rhetoric, and a narrative based on electoral fraud reinforced the image of the Republican candidate among populist politicians, the media, and citizens. Other far-right populist leaders placed him as the center of the conversation on Twitter; meanwhile, the press gave a large amount of coverage to the process, and polls stated that a huge percentage of citizens continue to consider him the winner of the election.

On the one hand, this study aimed to verify the presence of propaganda mechanisms in Trump's speech on Twitter and to determine the extent to which their themes and strategies coincide with those of other extreme right-wing populist leaders. On the other hand, we analyzed how the US elections were presented on the front pages of the international press and investigated the information bias with regard to Trump and news about the electoral process.

Our study offers insightful findings on the use of propaganda mechanisms as a common trend in the accounts of populist leaders on Twitter. These rhetorical resources reinforce both the issues (issue frame) and the strategies (game frame) used by politicians to promote polarization, attack opponents, and confusion of public opinion (Neudert and Marchal 2019). Beyond that, the legacy media are far from showing a critical attitude towards political lies, contributing to the development of different ideological approaches and increasing the level of disinformation that populism fosters (Bennett and Livingston 2018). Everything occurs in a context of platformization (Smyrnaios and Rebillard 2019), confrontation, and a political infodemic that affects the public sphere as a space for deliberative democracy (López-Borrull et al. 2018).

In response to **RQ1**, which was developed to investigate how the themes and strategies of far-right populism identify with the speech of Trump and the populist leaders on Twitter, a prevalence of the conflictive framework (horse race) as a game frame was detected. In addition to that, corruption was identified as the most relevant thematic issue in Trump's tweets. In this sense, the theory of fraud and delegitimization of the elections was the basis of his narrative, explaining the public belief about Trump's victory collected by the polls (Pew Research Center 2020). For European leaders (Salvini, Le Pen, and

Abascal), the most prominent frames were related to COVID-19 and immigration issues, topics avoided by Trump and Bolsonaro, given their controversial management of these problems.

Another of Trump's strategies shared by the rest of the leaders was the use of rhetoric on Twitter. The most commonly referenced resources consisted of appeal to emotion and the presentation of opinions as facts, showing a pattern of false messages. Information selection and the use of attributions also seem to be outstanding tools. Trump built false arguments about election fraud through the use of simple language, the selection of information, and by attacking other sources (adversaries, institutions, media, etc.). The use of spectacular language for propaganda purposes was used as a strategy by all leaders, regardless of geographic scope. This provided the answer to the second research question (**RQ2**), as that these mechanisms triggered the spread of disinformation. This is considered a characteristic of the messages of populist leaders on Twitter and endangers institutions and democracy.

Finally, we found that the press also reproduced messages of populism and disinformation on the digital front pages. In line with the discourse of the leaders on Twitter, the fallacy and mechanisms of propaganda were integrated into the news. However, there was less weight given to resources such as the appeal to emotion and the use of opinions as facts. Instead, other mechanisms such as attributions and information selection were used. It is remarkable that most of the front pages contained elements of the populist narrative.

Likewise, the information bias in news coverage was identified as a factor shared by the newspapers with a prevalence of 80% over the use of positive or neutral tones. As in other election contests held in the United States and other European contexts, the level of hostility between populist leaders and the media has been constant (Pérez-Curiel 2020). The negative tone used to describe the attitudes of the leaders, and a large amount of news on the American elections was identified as trends in all the analyzed newspapers. This dynamic collides with their responsibility as verifiers and guarantors of journalistic quality (Palau-Sampio 2018). In this sense, the third research question (**RQ3**) was also answered. The media were found to have an information bias in the coverage of issues published by populist leaders on Twitter.

Therefore, we argue that, like Trump, some of the main global populist leaders share speeches full of strategy and propaganda mechanisms on Twitter, especially messages containing emotion and the absence of factuality. Likewise, this study confirmed the negative bias and the prominence of disinformation on the US elections in the press, imitating their linguistic schemes. We showed that the leaders of far-right populism reproduced Trump's themes and strategies, reinforcing the idea of electoral fraud through mechanisms that promote disinformation. The international media also depicted the fallacy spread by politicians on Twitter on their front pages, revealing a significant critical attitude with a negative information bias in the coverage of facts and opinions.

Our findings are part of a wave of global illiberal populism, which has several characteristics (Waisbord and Amado 2017). This movement has implications in the public sphere, threatening the future of democracy (Moernaut et al. 2020). In the 2020 US elections, this was evidenced by the rejection of the results by many Republican voters (Pew Research Center 2020). However, our research also confirms that there is a certain level of adaptation of these strategies and fallacies depending on the national context, beyond an international trend with points in common.

A limitation of this article concerns the reduced volumes of messages on Twitter disseminated by leaders such as Abascal or Le Pen, in contrast to the levels of production of tweets by other politicians. However, the main objective of this study required us to focus on the US elections. It would be of interest to study other elections in which popular populist leaders participate in future studies as well as to evaluate their behavior during non-electoral periods. The impact of the elections in the United States gave relevance to the time frame studied, although broader longitudinal approximations could further our

understanding of how populist strategies enter the quality press. Additionally, academic works on fact-checking are relevant to this matter, highlighting the role of journalists as verifiers of fake news.

In conclusion, this contribution confirms the hybrid nature of populist communication and how it permeates the mainstream media from Twitter. This finding is relevant because the media selected have also been anti-right-wing populist press, advocating for cosmopolitan values. Besides that, the messages of the main international leaders are similar to Trump's speech during the US elections, prioritizing false and fraud-related content. In short, this communication model may spur cynicism and distrust towards democracy. It also fosters the negation of the electoral results and likely violent actions such as those witnessed later during the assault on the US Capitol. According to the Twitter messages analyzed, these trends are supported by extreme right-wing populism worldwide.

**Author Contributions:** Conceptualization, C.P.-C.; methodology, R.R.R. and M.G.-G.; investigation, C.P.-C., R.R.-d.-R. and M.G.-G.; resources, R.R.-d.-R.; writing—original draft preparation, C.P.-C and R.R.R.; writing—review and editing, M.G.-G.; supervision, C.P.-C. and M.G.-G.; project administration, C.P.-C. and R.R.-d.-R. All authors have read and agreed to the published version of the manuscript.

**Funding:** This research was funded by the VI Research Plan University of Seville [grant number IV.3-2017, Periodismo II].

**Data Availability Statement:** The data presented in this study are available on request from the corresponding author

**Acknowledgments:** Research team from the University of Seville that collaborated in the collection of the sample: Lucía Jiménez, Cristina Labrador, Francisco Javier Macías, María Martínez, and Berta.

**Conflicts of Interest:** The authors declare no conflict of interest.

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
