# Peer review of "Impact of Trump’s Digital Rhetoric on the US Elections: A View from Worldwide Far-Right Populism"

_socsci, doi:10.3390/socsci10050152_

Round 1
Reviewer 1 Report
This manuscript considers effects of Trump’s digital rhetoric on the US elections using Twitter data of five populist leaders and the front pages of the famous news papers. They set three objectives of this study as follows: to describe the impact of Trump’s speech on the five leaders, to analyze the issues (issue frame), strategies (game frame), and rhetorical marks of disinformation on Twitter and in the media, and to explore the information bias of the press. Using 1497 publications on Twitter and 112 front pages on digital press, they show that the leaders of far-right populism reproduced Trump’s themes and strategies, reinforcing the idea of electoral fraud through mechanisms that promote disinformation. Furthermore, they show that the international media depicted the fallacy spread by politicians on Twitter on their front pages, revealing a significant critical attitude with a negative information bias in the coverage of facts and opinions.
Despite of its careful survey on the related papers and their challenge to a practically significant topic, I regret my assessment that I can recommend neither this version nor a potential revised version as its publication. I have found some unignorable challenges in the current version.
First, they dealt with Twitter data of five populist leaders while they did not compare them with non-populist leaders. Even if we can ignore the definition of a populist, I think that their analysis never capture the features of the populist leaders. As well, they compare their Tweets with the legacy media while I think that its difference is not necessarily originated from the features of the populists. This methodological problem strikes the publication level essentially.
Second, did they analyze the data manually? For example, they categorized all twitter data into many labels when they made Tables 3 and 4. We do not know their methodological validity because they never mention it. A scientific rigorousness on their analyzing method is strongly requested.
Third, although I assess their message positively, their manuscript seems to be an essay. I cannot understand that they use what type of a research framework. If this manuscript is an article of a newspaper, I much appreciate their investigation. However, if it is for an academic paper, they must mention what they originally show. Its persuasion may including a suitable stastical analysis, of course.
Author Response
Thank you for your comments. The whole analysis was carried out manually by the authors, so we add now that information on Lines 293. Regarding the intercoder agreement, we provide the reach error level and additional details of our statistical research on Lines 293-299.
The rest of the changes requested are not applicable for this sort of research.
Reviewer 2 Report
The authors use a content analysis of tweets and newspapers to investigate the right-wing populist narrative of Donald Trump and other populist leaders in the aftermath of the 2020 US election. They find that there are similar frames and rhetorical elements in the online communication of right-wing populist politicians and that they were in line with Donald Trumps narrative.
While the text is well-written and offers a valuable insight in the global ideological network of populist politicians, there are some issues I would request the authors to take care of:
The first and biggest issue is the strange and unstructured assembly of hypotheses, objectives, and research questions:
- Hypothesis 1 is not really a hypothesis. There is no way to test whether other leaders "designed mechanisms" to reinforce Trump's narrative. And there is no null-hypothesis that may be assumed to conduct any kind of test. It might be better to formulate an open research question about the narratives used by the other leaders.
- In the wording of Hypothesis 2, there is a conceptual mistake. You write that the press uses narratives similar to right-wing populism "despite" having a critical bias to politicians. That "despite" is not accurate as this critical stance toward the political elite is the most salient and important cornerstone of populist communication. You should put an "especially", "specifically by", or just "by" instead of the "despite".
- The three research questions on Line 162-167 are quite surprising, given that you have already introduced hypotheses AND research objectives. In contrast to the hypotheses and objectives, these research questions do actually make sense, given your theoretical background. You should probably reformulate the hypotheses and objectives to more general intentions of the study (without enumeration) and stick to the RQs. After all, there is no mention of hypotheses and objectives in the discussion. The research questions, however, are addressed.
The second, and also quite important issue is regarding the sources of your bold claims in the theoretical background:
- The statement "In short, populism is strengthened on the world map due to the lack of trust of voters in governments and the media, the loss of reputation of the political establishment, and the decline of traditional parties." on Lines 92-94 is quite bold and not really backed by the literature you cite above. Of course, distrust in politics creates an opportunity structure for populism, but this explanation is not quite credible as is. Do you have any source backing this claim? At this point you need to back up your claims by a study in political theory. As a reader I would definitely need references to more detailed and well-founded information.
- Line 120 "Interestingly, Twitter also boosts fake news and the spreading of hate speech" again, a very bold statement that really needs to be backed by current research (there should be ample). This can't be taken at face value.
The third issue is methodological. For the quantitative content analysis, only a rudimentary coding scheme (table 2) is presented. There are no descriptions of the coders, their reliability, training, or control. Please add this information, if only in a footnote.
Minor points:
- Between lines 111 and 112, there is a giant cognitive leap. You jump from considerations on challenges to democracy to how Twitter is a social network. Either insert a blank line at this point to signify this leap or end the paragraph on line 111 with a first reference to social media networks. (e.g.: "These populist politicians have used social media channels like Twitter to criticize the legacy media..."). So, the reader is led gently through the argument.
- The selection of newspapers is justified in the paper solely by their circulation numbers. However, each newspaper is a strong advocate for left-wing policies and self-appointed enemy of far-right populists in their country. They always had skirmishes with right-wing parties and have become quite vocal about their opposition to right-wing populism in recent years. So, the selection is not just mainstream press but anti-right-wing-populist press. You should take this into account in the discussion.
Author Response
Firstly, we would like to thank your comments on the article, since the observations can help to improve the manuscript.
These are our responses to the first recommendation:
- As you stated, the hypothesis and objectives could be better defined. Hypotheses has been deleted both at the introduction and the discussion. Hypothesis 1 is formulated as an open research question about the populist narratives of the leaders (Lines 64-65).
- The conceptual mistake of Hypothesis 2 is solved. “Despite” is replaced by “specifically by”. Besides that, Hypothesis 2 is formulated through a reflection upon the role of the press (Lines 67-71).
- In order to simplify the structure of the article, we reformulate the objectives to more general intentions, deleting the enumeration (Lines 72-78). We just stick to the RQs as the main thread of the study.
Regarding the sources in the theoretical background, we include the following information:
- The statement “In short, populism is strengthened on the world map due to the lack of trust of voters in governments and the media, the loss of reputation of the political establishment, and the decline of traditional parties” has been changed, as it is true that was not really backed by the literature. Instead, we add that the “the raise of populist policies is fueled by a time of uncertainty and growing distrust in traditional politics” (Acemoglu et al. 2013). This new reference in Lines 104-105 is based on political theory: Acemoglu, Daron, Georgy Egorov, and Konstantin Sonin. 2013. A Political Theory of Populism. The Quarterly Journal of Economics 128(2): 771–805. https://doi.org/10.1093/qje/qjs077
- Lines 138-139 “Interestingly, Twitter also boosts fake news and the spreading of hate speech” is now backed by current literature (Bimber and Gil de Zúñiga 2020; Bracciale and Martella 2017). Please find here the whole data of these two new references:
-Bracciale, Roberto, and Antonio Martella. 2017. Define the populist political communication style: the case of Italian political leaders on Twitter. Information, Communication and Society 20(9): 1310–1329.
https://doi.org/10.1080/1369118X.2017.1328522
-Bimber, Bruce, and Homero Gil de Zúñiga. 2020. The unedited public sphere. New Media & Society 22(4): 700–715. https://doi.org/10.1177/1461444819893980
We are aware that the methodological design could be a limitation for this article, so an additional paragraph has been added on Lines 293-299 to reflect on this matter. The whole analysis was carried out manually by the authors. The intercoder agreement was calculated with Scott’s Pi formula, reaching an acceptable error level of 0.96. Control variables were not needed as our study did not aim to find casual relationships.
Minor points are also addressed:
- Following your recommendation to avoid the cognitive leap between Lines 111 and 112 (now Lines 124-126), there is a connecting sentence at the end of the paragraph: “These populist politicians have used social media channels like Twitter to criticize the contents of legacy media”.
- It is also true that we selected not only mainstream press but also anti-right-wing-populist press, so we include that point in the discussion (Line 534-542). The sentence is “This finding is relevant because the media selected have also been anti-right-wing populist press, advocating for cosmopolitan values”.
Reviewer 3 Report
Dear colleagues,
Congratulations for the interesting topic and approach.
I would like to point some remarks:
- Interesting number of tweets analyzed, and a mixed approach with the digital pages of global newspapers. The period of analysis is justified.
- In the abstract, line 16 (proclaimed winner by whom?)
- What is the link between the political leaders? Some of them belong to different European political families.
- The paper is focus on the impact of Trump in the communication of other leaders, however,it is important to note that the Le Pen phenomenon is before Trump`s victory.
- I would also recommend the distinction between populism with decision-making power and populism trying to assure parliamentary representation.
- Line 30-31 – the demonstration was made by radical republican supporters? Source...
- Line 50 – Jair (correction – name of Brazilian President).
- Objectives are perfect, according to the project.
- Focus on right-wing populism should deserve, in my opinion, a more accurate explanation.
- There are several relevant authors not mentioned in the methodological approach. I would recommend reviewing the selection of bibliography in this part (more traditional and central authors – ex. Bardin. (line 170)
- Table 2 is clear and is the key issue of the research and methodological approach.
- Good solution in the analysis of tweets (Salvini overrepresentation).
- The conclusion, debate and hypothesis validation are included in the same section. Normally they are divided. But it is mainly an option.
Author Response
Thank you for your remarks that were really helpful for our article. We take into account your proposals as follow:
- In the abstract (Lines 15-16), we include “the Democratic party candidate was proclaimed the winner of the elections by the media”, as in the US it is usually the press that calculates who is the winner according to the scrutiny.
- The link between the political leaders lies on its rhetoric. A paragraph in the introduction is included with that information (Lines 53-55): “Some of the leaders belong to different political families, but they have in common an aggressive rhetoric defending the interests of the ‘people’ against the elite” (Acemoglu et al. 2013).
- Since Le Pen phenomenon is before Trump’s victory, we note that matter in the text: “Indeed, Le Pen phenomenon happened before Trump’s victory in 2016, showing the early consolidation of populism in Europe” (Lines 100-102).
- As you stated, there is a distinction between populism in power and the one that tries to get parliamentary representation. To mention this, some new sentences are included (Lines 134-137): “However, populism with decision-making power is different from the populist leaders that try to achieve parliamentary representation. The aggressiveness of the rhetoric is higher in the latter, having both a permanent-campaign style” (Maatsch and Miklin 2021).
- Maatsch, Aleksandra, and Eric Miklin. 2021. Representative Democracy in Danger? The Impact of Populist Parties in Government on the Powers and Practices of National Parliaments. Parliamentary Affairs, Ahead of print.
- The demonstration was made by “extremist groups linked to the Republican party (The New York Times 2021)” (Line 32). The statement is now backed by this reference:
-The New York Times. 2021. Republican Ties to Extremist Groups Are Under Scrutiny. Available online: https://www.nytimes.com/2021/01/29/us/republicans-trump-capitol-riot.html (accessed on 20 April 2021).
- The name of Brazilian president has been corrected (Line 51).
- In order to provide an explanation of the research interest in the right-wing populist, this sentence is added at the end of the introduction (Lines 76-78): “The focus on right-ring populism is based on its huge use of social media and the impact on the public sphere” (Bimber and Gil de Zúñiga 2020).
- We reviewed the selection of bibliography in the methodological part, omitting some of them in this section (Pérez-Curiel 2020; Rivas-de-Roca et al. 2020) and giving more traditional authors: Bardin (1977) for the content analysis (Line 190), and van Dijk (2008) for the discursive analysis (Line 270):
-Bardin, Laurence. 1977. The Content Analysis. Paris: PUF.
-van Dijk, Teun A. 2008. Discourse and Context. A sociocognitive approach. Cambridge: Cambridge University Press.
- The conclusion, debate and research questions are present in the same section, since we consider that this approach is useful to better understand this research as a whole. Hypotheses were deleted and transformed into general intentions of the study following the advice of reviewer 2.